# Exploring the Role of Platelets in Virus-Induced Inflammatory Demyelinating Disease and Myocarditis

**DOI:** 10.3390/ijms25063460

**Published:** 2024-03-19

**Authors:** Ijaz Ahmad, Seiichi Omura, Fumitaka Sato, Ah-Mee Park, Sundar Khadka, Felicity N. E. Gavins, Hiroki Tanaka, Motoko Y. Kimura, Ikuo Tsunoda

**Affiliations:** 1Department of Microbiology, Faculty of Medicine, Kindai University, 377-2 Ohnohigashi, Osakasayama, Osaka 589-8511, Japan; ijazahmad383@gmail.com (I.A.); somura@med.kindai.ac.jp (S.O.); fsato@med.kindai.ac.jp (F.S.); ampk@med.kindai.ac.jp (A.-M.P.); cls.sundar@gmail.com (S.K.); 2Department of Arts and Sciences, Faculty of Medicine, Kindai University, Osaka 589-8511, Japan; 3Department of Immunology, Duke University, Durham, NC 27708, USA; 4Department of Biosciences, Centre for Inflammation Research and Translational Medicine, College of Health and Life Sciences, Brunel University London, Uxbridge UB8 3PH, UK; felicity.gavins@brunel.ac.uk; 5Division of Tumor Pathology, Department of Pathology, Asahikawa Medical University, Asahikawa 078-8510, Japan; hiroki-t@asahikawa-med.ac.jp; 6Department of Experimental Immunology, Graduate School of Medicine, Chiba University, Chiba 263-8522, Japan; kimuramo@chiba-u.jp

**Keywords:** glycoprotein Ib α chain, bioinformatics analysis, RNA sequencing analyses, regulation of gene expression, dilated cardiomyopathy, neuroinflammatory disease, picornavirus infections

## Abstract

Theiler’s murine encephalomyelitis virus (TMEV) infection has been used as a mouse model for two virus-induced organ-specific immune-mediated diseases. TMEV-induced demyelinating disease (TMEV-IDD) in the central nervous system (CNS) is a chronic inflammatory disease with viral persistence and an animal model of multiple sclerosis (MS) in humans. TMEV infection can also cause acute myocarditis with viral replication and immune cell infiltration in the heart, leading to cardiac fibrosis. Since platelets have been reported to modulate immune responses, we aimed to determine the role of platelets in TMEV infection. In transcriptome analyses of platelets, distinct sets of immune-related genes, including major histocompatibility complex (MHC) class I, were up- or downregulated in TMEV-infected mice at different time points. We depleted platelets from TMEV-infected mice by injecting them with platelet-specific antibodies. The platelet-depleted mice had significantly fewer viral antigen-positive cells in the CNS. Platelet depletion reduced the severities of TMEV-IDD and myocarditis, although the pathology scores did not reach statistical significance. Immunologically, the platelet-depleted mice had an increase in interferon (IFN)-γ production with a higher anti-TMEV IgG2a/IgG1 ratio. Thus, platelets may play roles in TMEV infection, such as gene expression, viral clearance, and anti-viral antibody isotype responses.

## 1. Introduction

Viral infection can cause tissue damage directly by infected cell lysis (viral pathology) or indirectly by the host’s immune responses to the pathogen (immunopathology) [1]. Several virus-induced diseases have also been shown to result from both viral pathology and immunopathology to some extent. The tissue tropism of viruses has often determined the target organs in viral infections; most viruses have been shown to have an affinity for specific tissues and only infect their target organ. For example, poliovirus can selectively infect and damage motor neurons that express poliovirus receptors in the central nervous system (CNS). Coxsackievirus B (CVB) can infect the heart and induce viral myocarditis [2]. Severe acute respiratory syndrome coronavirus 2 (SARS-CoV-2) has been shown to infect not only the respiratory system but also the CNS and other general organs, including the heart, inducing CNS pathology and myocarditis [3].

Viral tropism and the pathogenesis of distinct organs have been studied using animal models with various viruses with different tropisms. Theiler’s murine encephalomyelitis virus (TMEV) is a non-enveloped, positive-sense, single-stranded RNA virus that belongs to the family *Picornaviridae* and genus *Cardiovirus*, and is a natural pathogen of mice [4,5,6]. Following the viral inoculation, TMEV can infect both the CNS andheart, causing CNS and cardiac inflammation [7] (Figure 1). Thus, TMEV infection is a unique experimental system to clarify organ-specific viral-induced inflammatory diseases.

Experimentally, TMEV infection has been used as a viral model of multiple sclerosis (MS), since TMEV can induce a chronic inflammatory demyelinating disease [TMEV-induced demyelinating disease (TMEV-IDD)] in the CNS of susceptible mouse strains, such as SJL/J and C3H mice [8]. Following intracerebral TMEV inoculation, TMEV initially infects the gray matter of the brain, inducing inflammation (Figure 1A). Later, around 1 month post infection (p.i.), TMEV persistently infects macrophages and glial cells, including oligodendrocytes and microglia [9], with the infiltration of lymphocytes and macrophages, inducing inflammatory demyelination in the white matter of the spinal cord (Figure 1C) [4,9]. Although anti-viral antibodies and CD4^+^ and CD8^+^ T cells have been shown to be essential in viral clearance, these immune effectors can also play pathogenic roles in demyelination [10]. Thus, TMEV-IDD is similar to MS immunologically and histologically. Although the precise pathomechanism of MS is still unknown, the viral etiology of MS has been supported clinically and epidemiologically. For example, several viruses, including human herpesvirus 6 (HHV-6), Epstein–Barr virus (EBV), and picornaviruses, have been detected in MS samples [11]. EBV has been identified as a potential environmental trigger of MS; immune responses to EBV were higher in MS patients than in a healthy population [12,13].

TMEV infection has also been used as a mouse model of viral myocarditis [14]. Myocarditis is an inflammatory disease of the heart [15,16] and can be caused by a variety of viruses, such as human immunodeficiency virus (HIV) and hepatitis C virus (HCV), in humans [17,18,19]. Picornaviruses, particularly CVB, are a well-known pathogen of myocarditis [20]. Viral myocarditis can be divided into three distinct phases. TMEV-induced myocarditis has been shown to develop all three phases: in phase I, acute viral infection (around 4 days p.i.), the virus directly infects and replicates in the cardiac muscles and damages cardiomyocytes (viral pathology). Activated innate immune responses against the virus are involved in the viral clearance and recruitment of immune cells. In phase II, anti-viral cellular and humoral responses are activated (around 1 week p.i.); anti-viral immune responses not only eradicate the virus but also damage cardiomyocytes (immunopathology). In phase III (around 1 month p.i.), when the cardiac tissue damage is severe in phases I and II, it results in cardiac fibrosis and remodeling without viral persistence in the heart [21]. Although the precise pathomechanism of human and experimental myocarditis is unclear, an immunomodulating therapy without suppressing anti-viral immune responses has been suggested for the treatment of viral myocarditis [22,23,24].

Previously, we conducted a transcriptome analysis using blood and heart samples from TMEV-infected mice to identify the phase-specific gene expressions during the disease course of myocarditis [14,24]. Unexpectedly, we found the upregulation of seven platelet-related genes, including pro-platelet basic protein (*Ppbp*), platelet factor 4 (*Pf4*), and platelet glycoprotein IX (*Gp9*), in blood of TMEV-infected mice in phase III [24,25] (Appendix A). On the other hand, in an autoimmune model of MS, experimental autoimmune encephalomyelitis (EAE), Mehmood et al. (2023) conducted data mining using the spleen transcriptome data from two EAE models with distinct pathomechanisms induced in SJL/J and A.SW mice [26]. They reported the enriched gene ontology “platelet alpha-granule” and the enriched pathway “platelet activation” in EAE mice sensitized with the myelin oligodendrocyte glycoprotein (MOG)_92–106_ peptide; the upregulated genes included platelet-derived growth factor alpha (*Pdgfra*), megakaryocyte and platelet inhibitory receptor G6b (*Mpig6b*) [27], *Pf4* [28], and phospholipase A2 group VII (*Pla2g7*) [26]. The transcriptome data used by Mehmood et al. were our own experimental data deposited in the Gene Expression Omnibus (GEO) previously (Accession number: GSE99300) [29]. Thus, we reanalyzed the data and confirmed that platelet-related genes were upregulated in the spleen of two EAE models: 11 genes in SJL/J mice and five genes in A.SW mice. Intriguingly, although two EAE models had a primary progressive disease course clinically, only *Pdgfra* and *Pla2g7* were commonly upregulated in the two models (Appendix A). When we compared the seven upregulated genes in TMEV infection, the SJL/J EAE model had three genes, 1) multimerin 1 (*Mmrn1*) [30], 2) *Mpig6b*, and 3) *Pf4*, commonly upregulated, although the A.SW EAE model had no similarity with the seven upregulated genes in TMEV infection (Appendix A). Since platelets have been detected in lesions in EAE and MS previously [31], in this study, we focused on the role of platelets in TMEV infection.

Although platelets are known for their main roles in hemostasis and thrombosis, platelets can contribute to virus-induced inflammatory diseases by (1) binding virions and (2) modulating immune responses. First, platelets have been shown to express various viral receptors, facilitating direct interactions with viruses, such as influenza virus, CVB, and SARS-CoV-2 [32,33,34]. Several viruses have been reported to indirectly interact with platelets by the formation of virus and immunoglobulin (Ig) complexes, which are recognized by Fcγ receptor IIA on platelets [32]. Although platelet binding to viruses can lead to viral clearance [35], several viruses, including HCV and CVB, have been shown to use platelets as a vehicle for viral dissemination or replication within platelets [36,37,38,39].

Second, platelets have been shown to play various immunological roles in inflammatory diseases. For example, activated platelets can reach the sites of infection where they can modulate immune processes via expressions of immune molecules on their membrane, such as major histocompatibility complex (MHC) class I and CD40 ligand [40]. Platelets have been shown to contain various mRNAs, translate several proteins, and transport various inflammatory proteins or mRNAs, including chemokines and cytokines, to inflammatory sites [41,42,43,44]. Anti-platelet treatment, such as the administration of platelet-depletion antibodies, has been shown to modulate virus-induced inflammation experimentally [45,46].

In this study, we aimed to determine the role of platelets in the pathogenesis of TMEV-IDD and TMEV-induced myocarditis. Using RNA sequencing (RNA-seq), we found that distinct sets of immune-related genes, including MHC class I, were up- or downregulated in platelets at different time points; TMEV genomes were not detectable in the platelets at any time point. Platelet depletion in vivo in TMEV infection reduced viral persistence in the CNS and ameliorated histological severities by reducing inflammatory lesions in the spinal cord and fibrotic areas in the heart of TMEV-infected mice. Platelet depletion also increased interferon (IFN)-γ production and altered anti-viral IgG isotype responses, which could play a role in reduced viral persistence. Thus, platelets could be a new therapeutic target for MS and viral myocarditis.

## 2. Results

### 2.1. Expressions of Immune Genes Are Altered in the Spleen of TMEV-Infected Mice

During the first week of TMEV infection, viremia and TMEV-specific immune responses are observed. Then, innate and acquired immune responses eradicate the virus from the subject’s general organs by 2 weeks p.i., although the virus can persistently infect the spinal cord. We isolated mRNA from the spleen and platelets from TMEV-infected and control mice 4, 7, and 35 days p.i. By computational analyses of RNA-seq data, we compared the transcriptomes among the samples.

First, we conducted a two-way comparison analysis of splenic gene expressions between TMEV-infected and control mice at each time point (Figure 2A–C). In the spleen, heat maps showed the upregulations of innate immune genes, including 2’,5’-oligoadenylate synthetase (*Oas*) and interferon-induced protein with tetratricopeptide repeats (*Ifit*), 4 and 7 days p.i. as well as acquired immune genes, particularly Igs, 7 days p.i. Several other acquired immune genes, such as Igs and T cell receptors (TCRs), were upregulated 35 days p.i.; the transcriptome data of several immune molecules, including granzyme B, C-X-C motif chemokine ligand 10 (*Cxcl10*), the IgA heavy chain (*Igha*), and the IgG2c heavy chain (*Ighg2c*), were validated by real-time PCR testing (Appendix A). Using the Database for Annotation, Visualization, and Integrated Discovery (DAVID), we identified the functional clusters of differentially expressed genes (DEGs) in TMEV infection (Appendix A). A total of 80 and 135 genes were up- and downregulated 4 days p.i., respectively; 230 and 41 genes were up- and downregulated 7 days p.i., respectively; and 18 and 267 genes were up- and downregulated 35 days p.i., respectively. As expected, the DAVID functional clustering of DEGs showed that the immune system process-related pathways were activated 4 and 7 days p.i.

To further clarify the time course patterns of gene clusters, we conducted *k*-means clustering and divided the genes into 15 clusters based on the gene expression patterns (Appendix A; Figure 2D); an appropriate number of clusters was determined based on the Davies–Bouldin indexes (Appendix A). We examined the function of genes in each cluster with the DAVID (Appendix A). In a radar chart of the fifteen clusters, four clusters (clusters 1, 7, 10, and 12) changed the expressions at only one time point. Eleven clusters included genes whose expressions were changed at two or three time points. For example, cluster 6, composed of innate immune genes, including IFN-induced genes, was upregulated 4 and 7 days p.i.; cluster 14, consisting of various Ig and TCR genes, was upregulated 7 days p.i.; and cluster 15, composed of a smaller number of distinct Ig and TCR genes, was upregulated 35 days p.i. (Appendix A, Figure 2D,E).

### 2.2. Distinct Gene Expressions, Including Immune-Related Genes, in Platelets of TMEV-Infected Mice

Using the platelet samples from the TMEV-infected and control mice, we also conducted transcriptome analyses at the three time points of TMEV infection. Although we examined whether TMEV genomes were detected from the platelet transcriptome data of the TMEV-infected mice (Appendix A), no viral genomes were detected in the platelets at any time point. Our heat maps showed up- and downregulations of distinct genes at each time point (Figure 3A–C), including changes in immune-related genes (e.g., the upregulation of *H-2D1* 4 and 7 days p.i. and the downregulation of *Cd55* and *Cd52* 35 days p.i.). A total of 51 and 35 genes were up- and downregulated 4 days p.i., respectively; 92 and 39 genes were up- and downregulated 7 days p.i., respectively; and 130 and 209 genes were up- and downregulated 35 days p.i., respectively. Although we attempted to identify the genes in the functional pathways using the DAVID, the enrichment scores of identified pathways were lower than those in the spleen samples; immune-related pathways were not ranked among the top five pathways (Appendix A).

Using *k*-means clustering and the DAVID, we classified the genes into nine clusters (Appendix A; Figure 3D). MHC class I (*H-2D1*, *H-2K1*, and several *H-2Q* genes) [47] and β_2_ microglobulin (*B2m*) genes were upregulated in cluster 9 4 days p.i. Some innate immune genes, such as IFN-activated genes, were upregulated in cluster 3 7 days p.i. Microtubule-related genes, such as tubulin beta 2B class IIB (*Tubb2b*) and dynein axonemal heavy chains (*Dnah*), which are associated with the morphology and function of platelets [48], were highly upregulated in cluster 7 35 days p.i. Immune-related genes, including *Cd55*, *Itga8* [49], *Cd52*, and *Il16* [50], were downregulated in cluster 4.

### 2.3. Overall Gene Expression Profiles Differ in the Spleens, but Not in the Platelets between TMEV-Infected and Control Mice

Using a principal component analysis (PCA) of transcriptome data, we investigated whether the overall gene expression profiles differed between the TMEV-infected and control mice at each time point. In the spleen, the samples were separated significantly by principal component (PC) 2 values between the TMEV-infected and control groups 7 and 35 days p.i., but not 4 days p.i. (Figure 4A). A PCA using the splenic transcriptomes of the TMEV-infected mice separated the samples at each time point into three distinct groups (Appendix A). The PC1 values 35 days p.i. were significantly higher than those 4 and 7 days p.i. (*, *p* < 0.05, ANOVA, Appendix A); the PC2 values 7 days p.i. were significantly lower than those 4 and 35 days p.i. (***, *p* < 0.001, ANOVA, Appendix A). Factor loading for PC1 and PC2 showed that genes contributing to the PC1 and PC2 values included various immune-related molecules, such as Ig genes, reflecting changes in anti-viral immune responses at each time point (Appendix A). On the other hand, the PCA of the splenic transcriptome data of the control mice did not separate the groups of samples at each time point (Appendix A).

In the platelets, the PCA of the transcriptome data did not separate the samples between the TMEV-infected and control groups at any time point (Figure 4B). Thus, the individual gene expression changes (Figure 3) in TMEV infection were not significant enough to alter the overall transcriptome profiles (Figure 4B) of the platelets at any time point. On the other hand, when we conducted a PCA using the transcriptome data from the three time points, the PCA separated the samples into two groups: one group was composed of the samples 4 and 7 days p.i., and the other was composed of the samples 35 days p.i. (Figure 5A,B). We also conducted a PCA using all the TMEV-infected and control platelet transcriptome data; we did not see separations between the TMEV-infected and control samples (Appendix A). We found that the overall gene expression patterns of the samples from days 4 and 7 versus those of day 35 differed, which likely reflected the maturation-related gene expression changes in the platelets.

### 2.4. Platelets Are Present in the CNS and Heart of TMEV-Infected Mice

To determine whether platelets could play a role in the CNS and cardiac pathologies in TMEV infection, we killed the TMEV-infected mice 4, 7, 21, and 35 days p.i. and visualized the platelets using immunohistochemistry against platelet-specific markers, CD42b [platelet glycoprotein Ib α chain (GPIbα)] and CD61 (integrin β3) [51] (Figure 6). We also immunostained uninfected normal tissue sections as well as inflammatory CNS demyelinating lesions of mice with EAE, an autoimmune model of MS, in which platelet accumulations in the CNS have been reported previously [31]. In uninfected mice, we detected few or no platelets in the CNS and a small number of platelets in the heart (Figure 6A,C), since we perfused the mice extensively, removing their blood when we made histology sections. As a positive control, we used platelet-producing megakaryocytes (Appendix A). In EAE induced with the myelin proteolipid protein (PLP)_139–151_ or MOG_35–55_ peptide, we found a small number of platelets in the luminal side of the vascular endothelia, but not in the parenchyma adjacent to CNS inflammatory lesions (Appendix A).

In the CNS of TMEV-infected mice, we detected a small number of platelets sporadically, attached to the luminal side of the vascular endothelia in the inflammatory brain lesions 4 and 7 days p.i. (Appendix A, Figure 6B), and a much smaller number of platelets attached to the vessels in the spinal cord 21 and 35 days p.i. (Appendix A). In the heart of the TMEV-infected mice, we detected more substantial platelet accumulations than in their CNS; the platelets were observed diffusely, independent of cardiac lesions. The platelet accumulations in the heart peaked 7 days p.i. (Figure 6D), although a lower number of platelets was detected 4 days and 3 weeks p.i. (Appendix A–I) [14]. Thus, in the CNS and heart of the TMEV-infected mice, we found platelets adjacent to inflammatory lesions, although we did not see a parenchymal infiltration of platelets in either their CNS or heart.

### 2.5. Platelets’ Depletion Reduces Viral Persistence in TMEV-IDD

Using an antibody against GPIbα, we determined whether platelet depletion could alter TMEV-IDD histologically. We depleted the platelets by injecting the mice with an anti-GPIbα antibody; we found that a single injection of the anti-GPIbα antibody depleted 98.6% of the platelets in four days (Appendix A). To investigate the role of platelets in TMEV-infected mice, we injected the anti-GPIbα-antibody intravenously (i.v.) 0 and 5 days p.i. (early group) or 18 and 22 days p.i. (late group). The control group received the control IgG antibody. We killed the TMEV-infected mice 35 days p.i. and compared the number and distribution of viral antigen-positive (antigen^+^) cells by immunohistochemistry with an anti-TMEV antibody among the three groups. In all groups, the viral antigen^+^ cells were mainly detected in the white matter of the ventral and lateral funiculi of the spinal cord with minimal involvement of the dorsal funiculus, as reported previously in TMEV-IDD [52] (Figure 7A). We detected a significantly smaller total number of viral antigen^+^ cells in the early and late groups than in the control group (Figure 7B, overall, *, *p <* 0.05 and **, *p <* 0.01, ANOVA). In the ventral funiculus, viral antigen+ cells were significantly smaller in the early group (*, *p <* 0.05, ANOVA); in the lateral funiculi, there were significantly fewer in the early and late groups than in the control group (*, *p <* 0.05, ANOVA). On the other hand, there were few viral antigen+ cells in the dorsal funiculus, and there were no statistical differences among all the groups (Figure 7B). Thus, although platelet depletion reduced the number of viral antigen^+^ cells, it did not change the spatial distribution of the viral antigen^+^ cells in the spinal cord. We also quantified the spinal cord pathology scores 35 days p.i. The levels of demyelination and overall pathology were lower in the early group than in the control group (*p* = 0.098, *p* = 0.088, ANOVA); there were no statistically differences between the late and control groups (Figure 7C). There were also no statistical differences in meningitis among all groups, although the late group had higher scores of perivascular cuffing compared with the early and control groups (*, *p* < 0.05, *p* = 0.053, ANOVA).

### 2.6. Platelets Contribute to the Pathogenesis of Myocarditis

The TMEV-infected mice had multiple focal fibrotic lesions with calcification in their hearts 35 days p.i.; only a small number of infiltrating immune cells were sporadically detectable (Figure 8A) as reported previously [14]. We used picrosirius red staining for visualizing collagens I and III (Figure 8B). We also examined the T cell infiltration in the heart with immunohistochemistry against CD3 (Figure 8C). Our quantification of the fibrotic areas showed that the early and late groups had smaller lesion areas than the control group (Figure 8D). Although only a small number of T cells were sporadically detectable in all groups; we detected more CD3^+^ T cells in the early group (**, *p* < 0.01, ANOVA) than in the control and late groups (Figure 8E).

### 2.7. Platelet Depletion Alters Antibody Isotype Responses

We collected the sera from the TMEV-infected mice 35 days p.i. and compared antiviral IgG isotype titers among the three groups, using enzyme-linked immunosorbent assays (ELISAs). We found that all the groups had substantially high amounts of anti-TMEV total IgG titers, although the late group had a significantly lower amount of anti-TMEV total IgG titers (**, *p* < 0.01, ANOVA) compared with the early and control groups (Figure 9A). The number of anti-TMEV IgG1 titers was the least in the late group (Figure 9B); the number of anti-TMEV IgG2a titers was the highest in the early group (Figure 9C). We compared the IgG2a versus IgG1 ratios reflecting the T helper (Th) 1/Th2 balance among the three groups and found that the early and late groups had significantly higher ratios than the control group (**, *p* < 0.01, *, *p* < 0.05, ANOVA, Figure 9D).

### 2.8. Platelet Depletion Alters Lymphoproliferative Responses and Cytokine Production

In TMEV infection, the antiviral T cell responses can play either a beneficial role in viral clearance or a detrimental role in inflammatory demyelination [53]. We investigated whether platelet depletion could affect the lymphoproliferative responses to TMEV. We isolated mononuclear cells (MNCs) from the spleen of TMEV-infected mice and conducted lymphoproliferative assays using the Cell Counting Kit-8 (CCK-8). We found that the levels of anti-TMEV lymphoproliferative responses were comparable among the three groups (Appendix A).

We also quantified their cytokine production using MNC culture supernatants stimulated with TMEV or a mitogen by ELISAs. In TMEV stimulation, we compared the concentrations of four cytokines among the three groups. We detected higher levels of IFN-γ production in the early and late groups compared with the control group, although no statistical differences were seen among the three groups. In the TMEV stimulation, the amounts of interleukin (IL)-10 were similar among the three groups; those of IL-17 and IL-4 were not detectable in the three groups (Figure 10A,B). On the other hand, for the mitogen stimulation, we detected similar levels of IL-17, IFN-γ, IL-4, and IL-10 concentrations among the three groups (Figure 10C,D).

## 3. Discussion

Platelets can bind and internalize various virions, including HIV, for viral clearance [54] or, in turn, can be utilized by viruses, including SARS-CoV-2 and dengue virus [55,56], to spread to various organs. Platelets have also been shown to play a beneficial or detrimental role in viral infections in various organs, including the CNS and heart by modulating immune responses [57,58]. In the current study, we aimed to clarify the role of platelets in TMEV infection, which has been used as a viral model for immune-mediated CNS and cardiac diseases, i.e., MS and myocarditis. TMEV has been shown to infect most cell types in vitro [59] and can cause viremia in vivo following experimental inoculation [7]. However, in vivo, TMEV can replicate efficiently only in limited cell types and organs (i.e., heart, skeletal muscle, intestine, and CNS); TMEV was eradicated by anti-viral immune responses by 2 weeks p.i. from the most organs except the CNS. In the CNS, TMEV initially infected neurons in the gray matter of the brain; later, TMEV persistently infected macrophages and glial cells in the white matter of the spinal cord. Although the viral receptor of TMEV has not been identified [60,61], in theory, platelets could contribute to viral clearance or spread to the organs, if TMEV can bind platelets. However, this was not the case in TMEV infection, since we did not detect the viral genomes from platelets isolated from the TMEV-infected mice at any time point (Appendix A).

Besides virus–platelet binding, platelets have also been suggested to play diverse immunological roles, modulating viral infections and immune-mediated diseases of the CNS and heart, including MS [62,63,64], by various mechanisms. Platelets contain immune-related mRNAs and proteins and all molecular machinery, including ri-bosomes, to translate mRNA [65,66]. Thus, platelets can deliver, transfer, or express immune-related molecules/mRNAs from systemic circulation to various organs, interacting with various immune cells and vascular endothelial cells [67]. In TMEV infection, both humoral and cellular immune responses have been shown to contribute to viral clearance, although uncontrolled immune responses play a detrimental role, resulting in tissue damage in the CNS and heart. As reported in other viral infections, initially, TMEV infection induced innate immune responses followed by the induction of anti-viral T cell and antibody responses around 1 week p.i. and anti-viral antibodies and CD4^+^ and CD8^+^ T cells contributed to viral clearance in the brain and heart 2 weeks p.i. [68,69,70]. On the other hand, around 1 month p.i., the immune effectors play a detrimental role in TMEV-IDD or in fibrosis and cardiac remodeling in myocarditis.

In the current study, we conducted transcriptome analyses using spleen samples as a representative of lymphoid organs and found the upregulation of innate immune genes, including 2′-5′-oligoadenylate synthetases 2 (*Oas2*) and 1G (*Oas1g*) 4 and 7 days p.i., as well as distinct Ig- and TCR-related genes, including IgG heavy chain variable region (*Ighv*), *Ighg2c*, and *Igha*, 7 and 35 days p.i. (Figure 2). These results were consistent with the distinct roles of each immune effector during the time course of TMEV infection, as described previously [24,57,64,71]. In addition, using the DAVID functional clustering of DEGs in the spleen (Appendix A), we demonstrated that the enrichment scores of multiple immune-related pathways, including “immune system process,” “defense response,” “innate immune response,” and “antigen processing and presentation,” were high 4 and 7 days p.i. Furthermore, our PCA of the splenic transcriptome data showed that the overall gene expression profiles differed between the TMEV-infected and control mice (Figure 4A) as well as between the samples 4, 7, and 35 days p.i. (Appendix A).

Unlike the spleen samples, the platelet transcriptome analyses using the DAVID showed that the enrichment scores of immune-related pathways were low (Appendix A); our PCA of the platelet transcriptomes did not separate the samples between the TMEV-infected versus control groups (Figure 4B), but separated the samples on days 4 and 7 versus the samples on day 35 (Figure 5A,C). Thus, in the platelets, immune-related gene expressions changed individually, but not in specific immune pathways, which did not alter the overall gene expression patterns. The overall platelet gene expression profile seemed to be influenced by aging; previously, age-related changes in the expression levels of various platelet mRNAs, including immune-related genes, have been reported in both mice and humans [72,73,74,75,76,77,78,79].

Although we did not determine how these individual immune gene expressions were changed in the platelets of the TMEV-infected mice, the gene expression changes were unlikely to be caused by direct virus infection in the platelets, since we did not detect the viral genome in the platelets at any time point (Appendix A). We also did not determine the precise roles of individual platelet genes whose expressions changed over the course of TMEV infection. The following genes at each time point potentially could modulate anti-viral immunity as well as immunopathology: the upregulations of MHC class I-related genes and IFN-related genes 4 and 7 days p.i.; and the downregulations of distinct immune-related genes, including *Cd55*, 35 days p.i. For example, upregulated MHC class I (*H2-D1*, *H2-K1*) and β_2_ microglobulin (*B2m*) genes on platelets could result in the modulation of anti-viral CD8^+^ T cell responses, since platelets can present viral antigens to CD8^+^ T cells in infections [80,81]. CD8^+^ T cells have been shown to be protective, clearing virus-infected cells, or autoreactive, causing tissue damage in TMEV infection [82,83]. CD55, the decay-accelerating factor (DAF) complement, dissociates the C3 convertase, preventing the generation of C3a; the downregulation of CD55 could result in an exacerbation of inflammation. The regulation of inflammation by CD55 has been reported in viral infections and immune-mediated diseases [84,85]. On the other hand, we found the upregulation of microtubule-related genes in cluster 7, including tubulin and dynein, 35 days p.i. Microtubules and microtubule-binding proteins, such as dynein and kinesins, play key roles in platelet shape changes (discoid to spherical), activation, and granule exocytosis [86,87,88,89]. In Appendix A, we demonstrated the upregulations of the microtubule-related genes, including tubulin (*Tubb2b*), dyneins (*Dnah2*, *Dnah6*, *Dnah7a*, *Dnah9*, and *Dnah17*), and kinesins (*Kif1a*, *Kif1b, Kif9*, *Kif16b*, *Kif21a*, *Kif26b*, and *Kifc3*). Although we do not know the functional relevance of these molecules and their interactions in TMEV infection, platelet activation and migration have been reported to depend on a series of changes in their dynamic cytoskeleton and morphology; for example, spherical platelets are transported more quickly to the vessel wall than disc-shaped platelets, giving them a better chance to adhere near the inflammatory site.

Histologically, we detected platelets in the CNS and hearts of the TEMV-infected mice. In the CNS, we found a small number of platelets attached to the luminal side of the vascular endothelia, adjacent to inflammatory lesions. The extent and localization of platelets in TMEV infection were similar to those in EAE, whose platelet accumulation has been reported previously [31]. In the hearts, platelet accumulations were more diffuse, independent of focal lesions. Since we found that the platelets did not infiltrate into the parenchyma of the inflammatory lesions in the CNS or heart, the platelets seemed to be indirectly involved in TMEV-IDD and myocarditis by modulating immune responses in the peripheral lymphoid organs. Platelets could also contribute to an exacerbation of inflammation, in which platelets bound to the luminal side of inflamed vessels, serving as a kind of “platform,” recruit activated T cells, which could lead to an enhancement of T cell infiltration in the lesions [90].

The above platelet transcriptome analyses and histological detection of platelets in the CNS and heart suggested that platelets may contribute to viral clearance or immunopathology. Thus, we tested whether platelet depletion in TMEV infection could change the pathology of TMEV-IDD and myocarditis. We found that platelet depletion decreased the levels of viral persistence in the spinal cord (Figure 7A,B). Platelet depletion also ameliorated the neuropathology (demyelination and overall scores) in the spinal cord and fibrosis in the heart, although neither the CNS nor cardiac pathology scores reached a level of statistical significance compared with the controls (Figure 7C and Figure 8). Our findings were consistent with the previous reports that platelet depletion has been demonstrated to ameliorate other immune-mediated disease models, including EAE [31,91,92,93].

We tested whether the decreased TMEV persistence in the CNS could be due to changes in anti-viral immune responses [94]. We found that both the platelet-depleted and control groups developed comparable anti-viral humoral and cellular immune responses, i.e., total anti-TMEV IgG titers and TMEV-specific lymphoproliferative responses, respectively (Figure 8A; Appendix A). On the other hand, the platelet-depleted mice had the higher IgG2a/IgG1 ratios than the control mice. These changes in anti-TMEV IgG isotype responses were consistent with the higher production of IFN-γ (representative Th1 cytokine) [95] in the platelet-depleted groups than in the control group; Th1 and Th2 cytokines contribute to the isotype switching of IgG2a and IgG1 in mice, respectively [96]. In addition, IFN-γ itself has been shown to contribute to the clearance of various viruses, including TMEV [97,98,99,100,101]. In TMEV infection, IL-17-producing Th17 cells seemed to play a pathologic role, and IFN-γ-producing Th1 cells can be protective [4,9]. In contrast, in EAE models, both Th1 and Th17 cells have been demonstrated to be pathogenic, although there have been exceptions in some EAE models in which IFN-γ played a protective role [29].

Although our platelet depletion experiments suggested a potential detrimental role of platelets in TMEV-IDD and myocarditis, there were potential confounding findings in our experiments, which inevitably occurred in most platelet depletion/blocking experiments, in particular, bleeding. Since our anti-GPIbα antibody injections successfully depleted platelets, the platelet-depleted mice had severe anemia with low hemoglobin concentrations, reduced red blood cell counts, and fecal occult blood (Appendix A) [102]; their white blood cell counts were also decreased, although these did not reach a statistical difference. In addition, simultaneous platelet depletion and intracerebral viral inoculation resulted in significant body weight loss in the early group, in which 45% (5/9) of the mice died, likely due to brain hemorrhages. Thus, we were not able to refute that the reduced blood parameters as well as hemorrhage-associated stress might have affected the outcomes of our experiments. Moreover, our pilot experiments showed that more than two antibody injections per mouse often made platelet depletion less efficient, likely due to the generation of neutralizing anti-rat IgG, which meant only up to two antibody injections could be used to reproduce a substantial amount of platelet depletion. Thus, one should be cautious when evaluating the results of platelet depletion experiments similar to our experimental setting.

To address this concern, several platelet antagonists blocking the receptors on platelets can be used as candidates that inhibit platelet function without depleting platelets. For example, cangrelor is an inhibitor of platelet P2Y_12_ receptor [103,104,105]; MRS2179 is a platelet P2Y_1_ receptor antagonist [106,107]; and monoclonal antibody against platelet glycoprotein IIb/IIIa (GPIIb/IIIa) inhibits platelet aggregation [108], although these antagonists have been reported to increase the risk of bleeding. Rather than direct platelet inhibition at a single receptor, other anti-thrombotic agents, such as apyrase, have been reported to show the multimodal, indirect inhibition of platelets without increasing the risk of bleeding [109]. On the other hand, if a specific interaction between platelets and other cell types is proven to play a crucial role in the pathophysiology of MS and myocarditis in the future, blocking these interactions by the platelet receptor ligands on the non-platelet cells (e.g., vascular endothelia, T cells, or macrophages) [31] is another strategy to overcome these adverse events.

In conclusion, we demonstrated platelet involvement in TMEV infection; we found changes in platelet gene expressions as well as platelet attachment to the vessels adjacent to inflammatory lesions in the CNS and heart. Platelet depletion decreased viral persistence in the CNS, which was associated with altered anti-viral IgG isotype responses and IFN-γ production. Therefore, platelets can be a target in virus-induced CNS inflammatory diseases, possibly also in MS, as well as viral myocarditis.

## 4. Materials and Methods

### 4.1. Animal Experiments

We purchased four-week-old female SJL/J mice from Jackson Laboratory (Bar Harbor, ME, USA) and four-week-old male C3H/HeNJcl mice from CLEA Japan, Inc., (Tokyo, Japan). Mice were maintained under specific pathogen-free conditions in the animal care facility at Louisiana State University Health—Shreveport (LSUHS, Shreveport, LA, USA) or Kindai University Faculty of Medicine (Osaka, Japan). We inoculated five- to six-week-old mice intracerebrally (i.c) with 2 × 10^5^ plaque forming units (PFUs) of the Daniels (DA) strain of TMEV [110]. Control mice were injected with phosphate-buffered saline (PBS). For transcriptome analyses, mice were killed 4, 7, and 35 days p.i., and platelet and spleen samples were harvested from TMEV-infected and control mice. We used five to seven mice/group/time point. All experimental procedures were approved by the Institutional Animal Care and Use Committee of LSUHS and Kindai University Faculty of Medicine and performed according to the criteria outlined by the National Institutes of Health (NIH) [111].

### 4.2. Platelet Isolation

Mouse blood was collected in the syringes prefilled with 0.1 mL of a citrate–dextrose solution (ACD, Sigma-Aldrich, Co., St Louis, MO, USA) by cannulation to the carotid artery [112]. Collected blood was centrifuged at 118× *g* for 8 min to obtain platelet-rich plasma followed by platelet isolation with centrifugation at 735× *g* for 10 min. Platelets were resuspended with PBS (pH 7.37–7.38) and stored at −80 °C until used.

### 4.3. RNA-Seq

Total RNA was isolated from platelets and the spleen using the RNeasy Mini Kit (Qiagen, Valencia, CA, USA) and TRIzol Reagent (Thermo Fisher Scientific Inc., Waltham, MA, USA), according to the manufacturers’ instructions. DNase treatment was conducted using the RNase-Free DNase Set (Qiagen) as a part of RNA isolation. The purity of all samples was determined by the NanoDrop One/One^c^ Spectrophotometer (Thermo Fisher Scientific Inc.). RNA-seq was conducted by MR DNA (Shallowater, TX, USA) using a HiSeq system (Illumina, San Diego, CA, USA). Raw sequence (FASTQ) data were mapped by “Spliced Transcripts Alignment to a Reference (STAR)” and counted by the R version 4.2.3 [113] and the R packages, “GenomicAlignments” and “TxDb.Mmusculus.UCSC.mm10.ensGene.” The data were normalized by the R package “TCC (Tag Count Comparison)” based on the “differentially expressed gene elimination strategy (DEGES)” [114]. The normalized read count data were converted to logarithmic data for further analyses. The FASTQ files and read count data were deposited into the GEO at the National Center for Biotechnology Information (NCBI, Bethesda, MD, USA; accession no. GSE253385, https://www.ncbi.nlm.nih.gov/geo/query/acc.cgi?acc=GSE253385 (accessed on 15 March 2024)).

### 4.4. Bioinformatics Analyses

Heat maps were used to visualize the expression patterns of the top 20 up- and downregulated genes in the spleen or platelets of TMEV-infected mice 4, 7, and 35 days p.i., respectively. We calculated the logarithmic ratio (log ratio) of TMEV-infected mice compared with control mice from the logarithmic read count data and created the heat maps using the R packages “gplots” and “genefilter” [71]. PCA was conducted as an unsupervised analysis to determine the similarities and differences among the samples, using the R program “prcomp” [24,115]. We created PCA graphs with 80% confidential ellipses, using the R packages “dplyr” and “ggplot2”. Factor loadings were used to rank a set of genes contributing to the PCA distribution [116]. We conducted *k*-means clustering to categorize the genes based on their expression patterns throughout the disease’s course using the R package “cclust” [24]. Davies–Bouldin indexes were calculated in the *k*-means clustering process to decide the optimum number of clusters [117]. A radar chart was drawn using the expression patterns of cluster centers [71]. To determine differentially expressed genes, we uploaded a list of ensemble gene IDs that were differentially expressed with *p <* 0.05, more than 2-fold up- or downregulated between the control and TMEV-infected groups to the DAVID (https://david.ncifcrf.gov/ (accessed on 15 March 2024)). The enrichment score was calculated using the Fisher’s exact test, which compares the number of differentially expressed genes in the sample, matching with the total number of genes included in each canonical pathway.

### 4.5. Platelet Depletion In Vivo

To deplete platelets in vivo, we injected mice i.v. with an antibody against GPIbα (CD42b, #R300, Emfret Analytics, Eibelstadt, Germany) at a dose of 4 μg/g body weight. The control group received the control rat IgG (#C301, Emfret Analytics, Eibelstadt, Germany). The anti-GPIbα antibody depleted more than 95% of platelets in mice within 1 h after injection [118]. Antibody solutions were injected twice to each group on the following days p.i.: the early group, days 0 and 5; the late group, days 18 and 22; and the control group, days 0 and 5 or days 18 and 22. We monitored the body weight changes daily for 35 days.

### 4.6. Neuropathology and Cardiac Pathology

We killed mice 35 days p.i. with isoflurane (FUJIFILM Wako Pure Chemical Corporation, Osaka, Japan), harvested their sera, and perfused them with PBS extensively followed by a 4% paraformaldehyde (PFA, FUJIFILM Wako Pure Chemical Corporation) solution in PBS [119]. We harvested the CNS and heart and fixed them with a 4% PFA solution. The spinal cord was divided into 12 to 13 transverse segments and the heart was divided into six to seven transverse slabs. We embedded the tissues in paraffin and made 4-μm-thick sections using the HM 325 Rotary Microtome (Thermo Fisher Scientific Inc. Osaka, Japan). We stained the CNS sections with Luxol fast blue (Solvent Blue 38; MP Biomedicals, LLC, Irvine, CA, USA) for myelin visualization and conducted neuropathological scoring of the spinal cords. We divided each spinal cord segment into four quadrants: the ventral funiculus, dorsal funiculus, and two lateral funiculi. Each funiculus containing meningitis, perivascular cuffing (inflammation), or demyelination was given a score of 1 in that pathological class. The total number of positive quadrants for each pathological class was determined and then divided by the total number of quadrants present on the slide and multiplied by 100 to obtain the percentage of involvement for each pathological class. An overall pathology score was also determined by recording a positive score if any pathology was seen in the quadrant and presented as the percentage of involvement [120]. For cardiac pathology, the heart sections were stained with hematoxylin (Sakura Finetek Japan Co., Ltd. Tokyo, Japan) and eosin (Thermo Fisher Scientific Inc.) for inflammation and calcification or picrosirius red (ScyTek Laboratories, Inc., Logan, UT, USA) for fibrosis. We analyzed the sections stained with picrosirius red to quantify the fibrotic areas by ImageJ (ImageJ, Wayne Rasband and contributors, NIH, USA, http://imagej.nih.gov/ij)) with a color threshold tool and a measure tool. The fibrotic area (mm^2^) was used to calculate % fibrotic area, using the following formula:$$\%fibrotic\;area=\frac{fibrotic\;area}{fibrotic\;area+non\;fibrotic\;area}\times 100$$

### 4.7. Immunohistochemistry

We visualized platelets, TMEV antigen^+^ cells, and T cells by immunohistochemistry using anti-bodies against GPIbα/CD42b (100-fold dilution, Abcam, Tokyo, Japan) and integrin β3 [platelet glycoprotein IIIa (GPIIIa/CD61), 500-fold dilution, Cell Signaling Technology, Inc., Tokyo, Japan], anti-TMEV antibody [121], and anti-CD3 antibody (T cell marker, 120-fold dilution, Biocare Medical, Pacheco, CA, USA). For GPIbα and integrin β3 immunostaining, the CNS, heart, and spleen sections were pretreated with a 10 mM citrate buffer at a pH of 6.0 (Agilent Technologies Japan, Ltd., Tokyo, Japan) for 15 min at 95 °C using an MI-77 temperature-controllable microwave (Azumaya Medical Devices Inc., Tokyo, Japan) for antigen retrieval. For CD3 immunostaining, the CNS and heart sections were pretreated with a 10 mM citrate buffer at a pH of 6.0 (Agilent Technologies Japan, Ltd.) for 15 min at 120 °C using an autoclave [120]. The antibody/antigen complexes were visualized by a secondary antibody Histofine MAX-PO kit (Nichirei Biosciences Inc., Tokyo, Japan) with a 3,3′-diaminobenzidine (DAB, FUJIFILM Wako Pure Chemical Corporation) substrate solution. To quantify the levels of viral persistence in the spinal cord, we divided each spinal cord segment into four quadrants: the ventral funiculus, dorsal funiculus, and the two lateral funiculi, and we counted the number of TMEV antigen^+^ cells in each quadrant of the spinal cord under a light microscope using a 10× objective lens. We quantified TMEV antigen+ cells using 12 to 13 spinal cord segments per mouse. To quantify T cell infiltration of the heart, we counted CD3^+^ cells in all six to seven transverse slabs of the heart per mouse.

### 4.8. Anti-TMEV Isotype Antibody ELISAs

We collected blood from the heart 35 days p.i. and centrifuged the blood at 2775× *g* at 4 °C for 20 min. We used sera to assess the levels of anti-TMEV antibodies by ELISAs. We coated 96-well flat-bottom Nunc-Immuno plates (Thermo Fisher Scientific Inc.) with 10 μg/mL of purified TMEV antigens. After blocking the plates with 10% fetal bovine serum (FBS) and 0.2% Tween^TM^ 20 (Thermo Fisher Scientific Inc.) in PBS, we diluted the serum samples with the blocking solution by serial two-fold dilutions from 2^7^ to 2^28^, added the diluted serum samples to the plates, and incubated the plates at room temperature (RT) for 75 min. We washed the plates with a washing buffer consisting of 0.2% Tween^TM^ 20 in PBS and added horseradish peroxidase (HRP)-conjugated anti-mouse IgG (H+L) (2000-fold dilution, Thermo Fisher Scientific Inc.), anti-mouse IgG1 (4000-fold dilution, Thermo Fisher Scientific Inc.), or anti-mouse IgG2a (4000-fold dilution, Southern Biotechnology Associates, Inc., Birmingham, AL, USA) secondary antibodies to the plates. After 90 min of incubation, we detected the immunoreactive complexes using the BD OptEIA^TM^ TMB Substrate Reagent Set (BD Biosciences, San Jose, CA, USA) according to the manufacturer’s instruction. The absorbances were measured at 450 nm using the Synergy H1 Hybrid Multi-Mode Microplate Reader (Agilent Technologies, Inc., Santa Clara, CA, USA). The anti-TMEV antibody titers was determined as the highest reciprocal of the dilution that had an absorbance higher than the average plus two standard deviations of naïve serum samples at a dilution of 2^7^-fold.

### 4.9. Lymphoproliferative Responses and Cytokine ELISAs

We harvested the spleens from the early, late and control mice 35 days p.i. To make single-cell suspensions, we mashed the spleens on a metal mesh with 50 μm pores using the plunger of a 5 mL syringe and isolated MNCs using Histopaque^®^-1083 (Sigma-Aldrich, Co.). The splenic MNCs were cultured in RPMI-1640 medium (Sigma-Aldrich, Co.) supplemented with 10% FBS (Sigma-Aldrich, Co.), 2 mM L-glutamine (Sigma-Aldrich, Co.), 50 mM β-mercaptoethanol (FUJIFILM Wako Pure Chemical Corporation), and 1% antibiotics (Thermo Fisher Scientific Inc.) at 2 × 10^5^ cells/well in 96-well plates (Sumitomo Bakelite Co., Ltd., Tokyo, Japan). We incubated the MNCs with or without 1 µg purified TMEV antigens at 37 °C with 5% CO_2_ for 5 days. To assess the levels of TMEV-specific lymphoproliferative responses, we added 3 µL/well of a CCK-8 solution (Dojindo Laboratories, Kumamoto, Japan) in the cell culture system for the last 24 h. All cultures were performed in triplicate and the absorbances were measured at 450 nm using the Synergy H1 Hybrid Multi-Mode Microplate Reader. The data are expressed as stimulation indexes: (mean absorbance of the wells stimulated with TMEV antigen)/(mean absorbance of wells without stimulation). For cytokine ELISAs, we cultured the splenic MNCs at 8 × 10^6^ cells/well in 6-well plates, stimulated the splenic MNCs with 8 × 10^6^ PFUs of TMEV (multiplicity of infection of 1) or 10 μg/mL of ConA (Sigma-Aldrich, Co.), and incubated the plates at 37 °C with 5% CO_2_ for 2 days. The culture supernatants were collected and stored at −80 °C until examined. The concentrations of IL-17A (BioLegend, Inc., San Diego, CA, USA), IFN-γ, IL-4, and IL-10 (BD Biosciences, San Jose, CA, USA) were quantified in the culture in duplicate wells using ELISA kits, according to the manufacturers’ instructions [120].

### 4.10. Statistics

Statistical analyses were conducted using OrigenPro 2023 (OrigenLab, Corporation, Northampton, MA). We conducted Student’s *t*-test for two groups and analysis of variance (ANOVA) with Fisher’s post hoc LSD test for three or more groups. *p* < 0.05 was considered to indicate a significant difference.

## Figures and Tables

**Figure 1 ijms-25-03460-f001:**
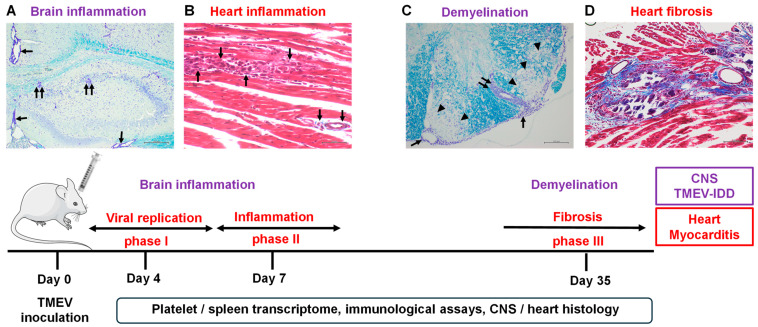
Schematic representation of the time course and experiments of Theiler’s murine encephalomyelitis virus (TMEV) infection. (**A**,**C**) In the central nervous system (CNS), following virus inoculation, TMEV initially infects neurons in the gray matter of the brain and causes inflammation (polioencephalitis). Later, around 1 month post infection (p.i.), TMEV persistently infects the white matter of the spinal cord and induces inflammatory demyelination. In Luxol fast blue stain, the arrowheads, arrows, and paired arrows indicate demyelination, meningitis, and perivascular cuffing (inflammation), respectively. Scale bars: (**A**) 200 μm; and (**C**) 100 μm. (**B**,**D**) TMEV-induced myocarditis can be divided into three phases. In phase I (around 4 days p.i.), TMEV directly attacks cardiomyocytes by infecting and replicating (viral pathology). In phase II (around 1 week p.i.), anti-viral T cell and antibody responses are induced, which not only clear the virus but also damage cardiomyocytes (immunopathology). In phase III (around 1 month p.i.), when the tissue damage caused in phases I and II is severe, it leads to cardiac fibrosis. In Masson’s trichrome stain, the arrows indicate cardiac inflammation; the blue and dark purple colors show fibrosis and calcification, respectively. Scale bars: (**B**) 50 μm; and (**D**) 100 μm. (Bottom) Experimental designs of this study. We intracerebrally inoculated mice with TMEV and harvested the spleen and platelet samples for transcriptome analyses, sera and lymphocytes for immunological assays, and the CNS and heart tissues for histological analyses 4, 7, and 35 days p.i.

**Figure 2 ijms-25-03460-f002:**
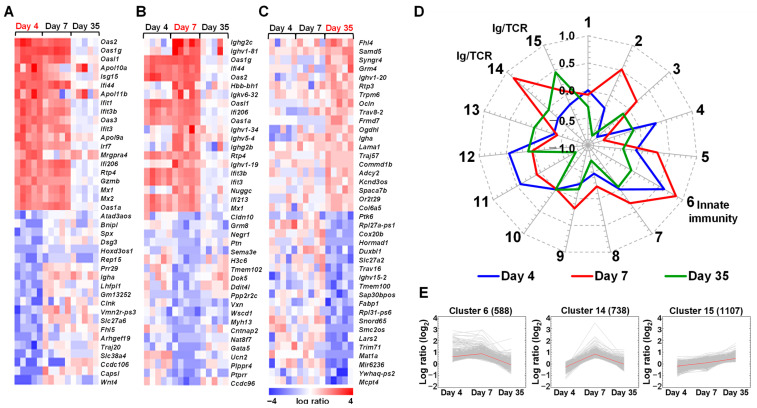
Transcriptome analyses of the spleen from TMEV-infected mice. (**A**–**C**) We created heat maps of the top 20 up- and downregulated genes 4 (**A**), 7 (**B**), and 35 (**C**) days p.i. The red, blue, and white colors indicate upregulation, downregulation, and no change, respectively, compared with the control samples. Each column represents the data from one mouse (five mice/time point). (**D**) We conducted *k*-means clustering and divided the genes into the 15 clusters based on their expression patterns. In a radar chart of the 15 cluster centers, radial axis values are fold-changes in the cluster center gene expressions compared with controls, which ranged from −1 to 1. These values are shown in binary logarithm (log_2_): −1, two-fold downregulation; 0, no change; and 1, two-fold upregulation, compared with controls. Cluster 6, composed of innate immune genes, including interferon (IFN)-induced genes, was upregulated 4 and 7 days p.i. Cluster 14, consisting of various immunoglobulin (Ig) and T cell receptor (TCR) genes, was upregulated 7 days p.i.; and cluster 15, composed of a smaller number of distinct Ig and TCR genes, was upregulated 35 days p.i. (**E**) Expression patterns of clusters 6, 14, and 15 showed the temporal gene expressions at each time point; the red lines indicate the cluster centers. The number of genes in each cluster is shown at the top of each cluster.

**Figure 3 ijms-25-03460-f003:**
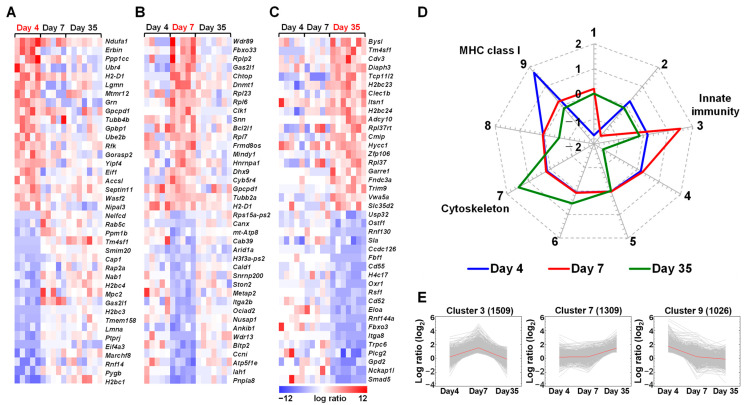
Transcriptome analyses of the platelets from TMEV-infected mice. (**A**–**C**) We created heat maps of the top 20 up- and downregulated genes 4 (**A**), 7 (**B**), and 35 (**C**) days p.i. The red, blue, and white colors indicate upregulation, downregulation, and no change, respectively, compared with the control samples. (**D**) We conducted *k*-means clustering and divided the genes into nine clusters based on their expression patterns. A radar chart using the nine cluster centers showed the distinct expression patterns of the platelet transcriptomes at each time point. Radial axis values are the cluster center gene changes compared with controls, which ranged from −2 to 2 and are shown in log_2_: −2, four-fold downregulation; 0, no change; and 2, four-fold upregulation, compared with controls. Genes in four clusters (clusters 4, 6, 7, and 8) changed their expressions at one time point. Genes in cluster 5 had few changes. Four clusters (clusters 1, 2, 3, and 9) included genes changed at two or three time points. (**E**) Expression patterns of three clusters (clusters 3, 7, and 9) showed the temporal gene expression patterns at each time point; the red lines indicate the cluster centers. Number of genes in each cluster is shown at the top of each cluster. Each time point was composed of five to seven mice.

**Figure 4 ijms-25-03460-f004:**
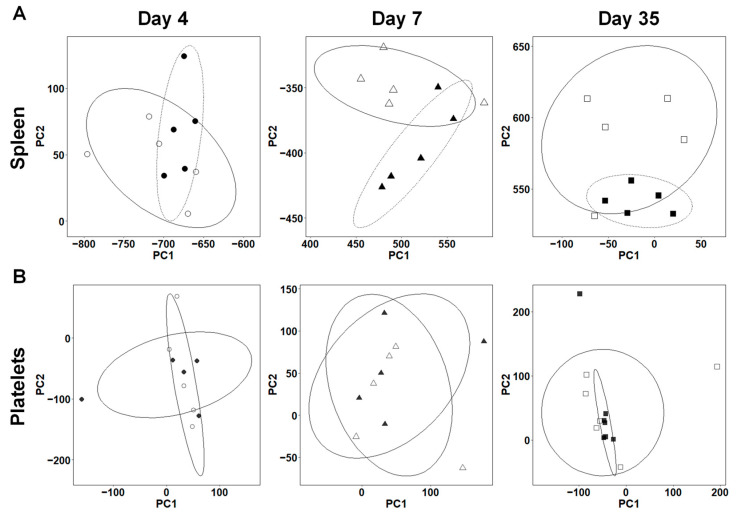
Principal component analysis (PCA) of splenic and platelet transcriptome data between TMEV-infected and control mice at each time point. (**A**) In the spleen, the samples were separated significantly by principal component (PC) 2 values between the TMEV-infected and control groups 7 days p.i.: TMEV (▲), −394.33 ± 14.32; control (△), −347.48 ± 7.95, *p* < 0.05, and 35 days p.i.: TMEV (■), 541.66 ± 4.30; control (□), 587.18 ± 15.10, *p* < 0.05, but not 4 days p.i.: TMEV (●), 68.42 ± 16.09; control (○), 46.06 ± 12.16, *p* = 0.348. (**B**) In platelets, the samples were not separated into distinct groups at any time point.

**Figure 5 ijms-25-03460-f005:**
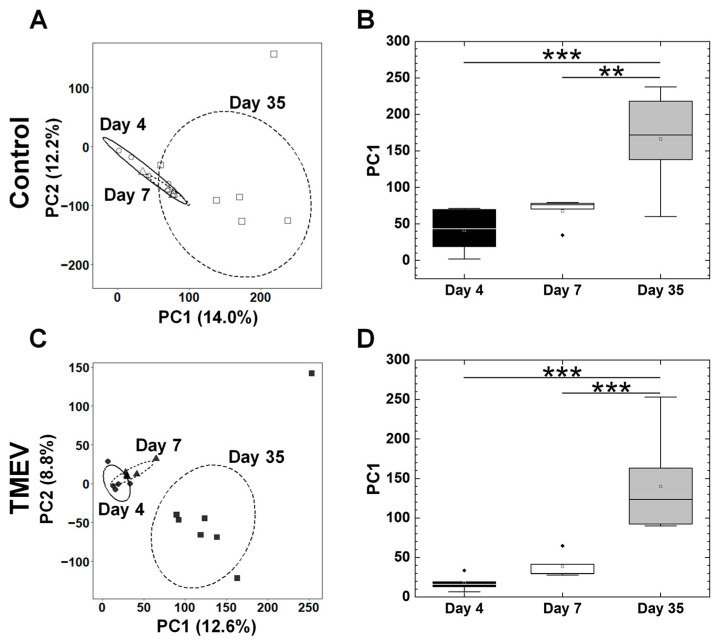
PCA of the platelet transcriptome data from TMEV-infected and control mice on day 4 (TMEV, ●; control, ○), day 7 (TMEV, ▲; control, △), and day 35 (TMEV, ■; control, □). (**A**,**B**) In the control samples, a PCA separated the samples into two groups: one group was composed of days 4 and 7 samples, and the other was composed of day 35 samples. The values in parenthesis indicated the proportion of variance of each PC. When the PC1 values were compared among the three time points, the PC1 values of day 35 samples were significantly different from those of days 4 and 7 samples (**, *p* < 0.01; ***, *p* < 0.001, ANOVA). (**C**,**D**) In the TMEV-infected samples, PCA and their PC1 values at each time point showed similar profiles to the control samples. In the boxplots: the open square, middle line, box, lower whisker, upper whisker, and closed rhombus indicate the mean, median, interquartile range, minimum, maximum, and outlier, respectively. The total sample number was five to seven per time point.

**Figure 6 ijms-25-03460-f006:**
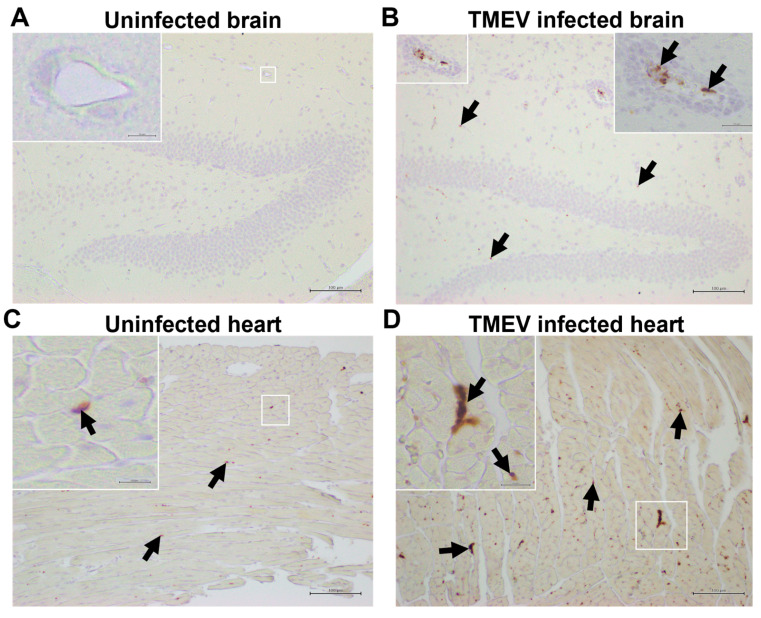
Platelet detection in the brain and heart of TMEV-infected mice. (**A**,**C**) We killed the uninfected mice and conducted immunohistochemistry against the platelet glycoprotein Ibα (GPIbα/CD42b) using brain and heart sections. We could not detect platelets in the brain of uninfected mice, although we detected a small number of platelets (arrows) in the heart. (**B**,**D**) We killed TMEV-infected mice 7 days p.i. and detected platelets attached to the luminal side of vascular endothelia in the brain and more diffusely in the heart. (**A**,**B**) hippocampus; and (**C**,**D**) heart**.** Scale bar: (**A**–**D**) 100 μm; and inset, 50 μm.

**Figure 7 ijms-25-03460-f007:**
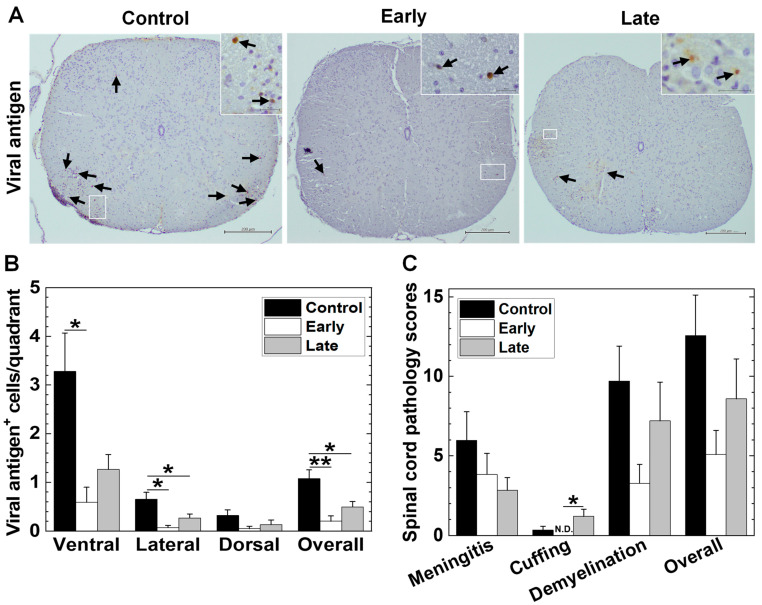
Immunohistochemistry against viral antigens in the spinal cord of TMEV-infected mice. We infected mice with TMEV, divided the mice into three groups, and intravenously injected them with a platelet-specific glycoprotein Ibα chain (GPIbα) depletion antibody 0 and 5 days p.i. (early group) or 18 and 22 days p.i. (late group), or with the control antibody (control group). (**A**) We found fewer viral antigen^+^ cells (arrows) in the spinal cords of the early and late groups than in the control group. (**B**) Overall, we found significantly fewer viral antigen^+^ cells in the early (**, *p* < 0.01, ANOVA) and late (** p* < 0.05, ANOVA) groups than the control group. In the ventral funiculus, we detected a smaller number of viral antigen^+^ cells in the early and late groups than in the control group. There was a statistical difference between the early and control groups (*, *p* < 0.05, ANOVA); the number of viral antigen^+^ cells tended to be lower in the late group than in the control group (*p* = 0.053, ANOVA). We detected significantly fewer viral antigen^+^ cells in the lateral funiculi of the early and late groups compared with the control group (*, *p* < 0.05, ANOVA). The spinal cord was divided into 12 to 13 transverse sections per mouse. (**C**) We quantified the pathological changes in the spinal cord using a spinal cord pathology scoring system. The levels of demyelination (*p* = 0.098, ANOVA) and overall pathology (*p* = 0.088, ANOVA) tended to be higher in the control group than in the early group, but not the late group. There were no differences in meningitis among the three groups, although the late group had higher scores of perivascular cuffing than the control (*p* = 0.053, ANOVA) and early (*, *p* < 0.05, ANOVA) groups. Each group was composed of five to fourteen mice. Results are the mean + standard error of the mean (SEM). Scale bar: (**A**), 200 μm; and inset, 20 μm. N.D., not detectable.

**Figure 8 ijms-25-03460-f008:**
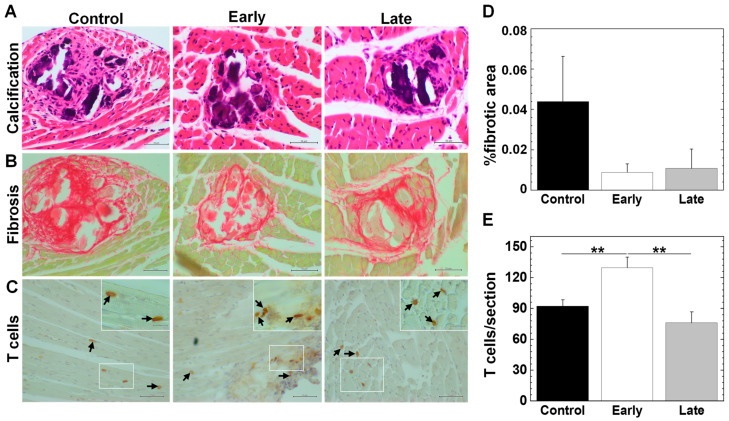
Cardiac pathology of TMEV-induced myocarditis. (**A**–**C**) We harvested the heart 35 days p.i. and dissected each heart into six to seven transverse sections. (**A**) Hematoxylin and eosin staining visualized calcification (dark purple). (**B**) Picrosirius red staining visualized fibrosis (red). (**C**) Immunohistochemistry against CD3 (T cell marker) showed CD3^+^ T cell infiltration in the heart (arrows). (**D**) Although the control group had larger fibrotic areas, there were no statistical differences among the three groups. (**E**) The number of CD3^+^ T cells in the heart was significantly higher in the early group than the control and late groups (**, *p* < 0.01, ANOVA). There were no statistical differences in CD3^+^ T cells in the heart between the control and late groups. (**D**,**E**) We quantified %fibrotic areas and the number of CD3^+^ T cells/heart section/mouse using ImageJ (version 1.53e). Values are the mean + SEM of five to fourteen mice per group. Scale bar: (**A**), 50 μm; (**B**), 50 μm; (**C**)**,** 50 μm; and inset, 20 μm.

**Figure 9 ijms-25-03460-f009:**
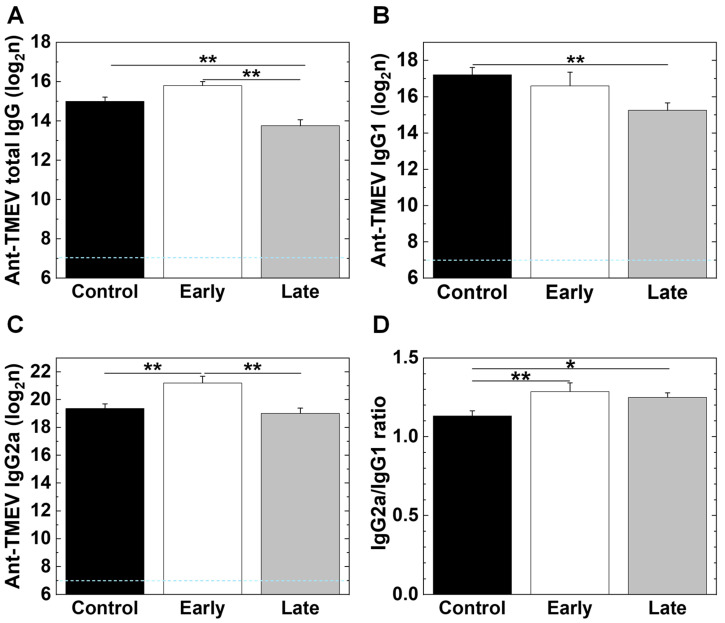
Anti-viral IgG isotype responses in TMEV-infected mice. We collected sera from the control, early, and late groups 35 days p.i. and compared anti-TMEV antibody titers (total IgG, IgG1, and IgG2a). The antibody titers were determined by enzyme-linked immunosorbent assays (ELISAs). (**A**) All groups had substantially high amounts of anti-TMEV IgG titers, although the control and early groups had significantly higher amounts of anti-TMEV total IgG titers (**, *p* < 0.01, ANOVA), compared with late group. (**B**) The number of anti-TMEV IgG1 titers was significantly lower in the late group than in the control group (**, *p* < 0.01, ANOVA). (**C**) Anti-TMEV IgG2a titers were significantly higher in the early group than in the late and control groups (**, *p* < 0.01, ANOVA). (**D**) The IgG2a versus IgG1 ratios, which reflect T helper (Th) 1/Th2 balance, were significantly higher in the early and late groups than in the control group (**, *p* < 0.01, *, *p* < 0.05, ANOVA). The IgG2a/IgG1 ratios were comparable between the early and late groups. Results are the mean + SEM of five to fourteen mice per group.

**Figure 10 ijms-25-03460-f010:**
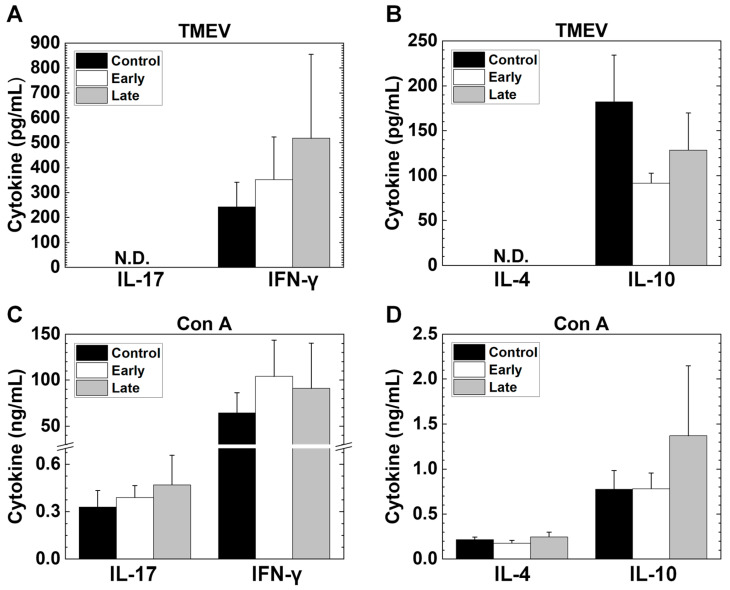
Cytokine production of splenic mononuclear cells (MNCs) from the control, early, and late groups. (**A**–**D**) Splenic MNCs were isolated from TMEV-infected mice and stimulated with TMEV (**A**,**B**) or a mitogen, concanavalin A (Con A) (**C**,**D**). The concentrations of interleukin (IL)-17, interferon (IFN)-γ, IL-4, and IL-10 in the culture supernatants were quantified by ELISAs. (**A**) There were no statistical differences in the concentrations of IFN-γ among the three groups, although the early and late groups had the higher levels of IFN-γ production than the control group in TMEV stimulation. (**B**) The concentrations of IL-10 were similar among the three groups. (**C**,**D**) The amounts of IL-17, IFN-γ, IL-4, and IL-10 were similar in response to Con A among the three groups. Results are the mean + SEM from two to six pools of spleens with two to three mice per group. N.D., not detectable.

## Data Availability

The data generated for this study can be found in the GEO at the NCBI (accession no. GSE253385, https://www.ncbi.nlm.nih.gov/geo/query/acc.cgi?acc=GSE253385 (accessed on 15 March 2024)).

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
