# Peer review of "Exploring the Role of Platelets in Virus-Induced Inflammatory Demyelinating Disease and Myocarditis"

_ijms, 2024, doi:10.3390/ijms25063460_

Round 1

Reviewer 1 Report

Comments and Suggestions for Authors

This is a very interesting study underlining the role of platelets not limited to hemostasis but also in inflammatory-immune reactions as important modulators, and the response to a viral infection is an other clinical setting of platelets involvement, as well demonstrated by the authors.

The only suggestion/comment is due to the main limitation of the study, addressed by the authors themselves: the bleeding for the depletion of platelets could interfere with the validation of in vivo results: did the authors tested, for in vivo experiments, other strategies as the administration of  platelets antagonists (cangrelor, MRS2179, or apyrase)? Did the authors consider any other strategies to overcome this important issue? An additional comment could be useful. 

Reviewer 2 Report

Comments and Suggestions for Authors

In this study, the authors aimed to elucidate the involvement of platelets in regulating immune responses within the TMEV-IDD mouse model of multiple sclerosis (MS), employing transcriptome and histopathological analyses. The findings are effectively presented; however, several aspects could be refined to enhance the manuscript and strengthen the scientific rationale.

Major comment:

The current results do not support a strong role for platelets neither in the pathology of spinal cord nor in the heart. For the spinal cord, as only one pathology score parameter was altered which is cuffing and not demyelination or meningitis, no change in platelet gene expression profile in response to TMEV infection was observed. As for the heart, altered parameters such as size of fibrotic areas (Fig 8D) and cytokine production (Fig 10) did not reach statistical significance. However, negative or sub-significant results are still important and add to the body of knowledge, but to be cautious the authors are advised to change the study title to, for example (exploring the role of platelets in Virus-Induced Inflammatory Demyelinating Disease and Myocarditis) and consider revising any assertive massages suggesting the involvement of platelets in these pathologies.

Specific comments:

1)       The findings from the transcriptome analysis are not validated by an independent method, can the authors explain why?

2)       Where there a subset of genes that were dysregulated across more than one time point?

3)       The authors mention that cytoskeleton-related genes were upregulated in the platelets of TMEV-infected mice 35 day p.i., can they elucidate in the discussion in what way can this subset of genes be involved in platelet response to viral infection ,for example does it modify the cell shape and motility to enable them reach the infection sites?

4)       TMEV infection did not demonstrate a significant alteration in platelets gene expression profile and the observed alteration across the selected time points, reflect maturation related process, did the authors re-run PCA on the combined time points on platelet transcriptome from TMEV infected vs control to see if there is an overall effect?

5)       In the results section (2.2) the authors stated that viral genome was absent from the platelets but present in the spinal cord, what does this observation indicate? Could the absence of viral genome in the platelets explain the lack of DEG profile?

6)        Why did the authors chose to present images only from 7 days p.i. in Figure 6? Is it because platelet accumulation was more prominent at this time point?

7)       In section 2.4 the authors state “ Thus, both in the CNS and heart, we found

platelets adjacent to inflammatory lesions, although we did not see parenchymal infiltration of platelets neither in the CNS nor the heart.” Could the authors emphasize that this observation applies to TMEV-infected only and not EAE model for clarity.

8)       Can the authors explain if there is a significance or implication of the differential spatial distribution of viral antigen+ cells between ventral, dorsal or lateral sides of the spinal cord.

9)       Can the authors include description of the methodology used to measure the size of viral antigen+ cells

Minor: comments

1)       What does the grey boxes in (figures S2 D,E)  signify?

2)       FigureS4,G the panel title “Hear” is missing the letter (t)

3)       Figure S4 (B,C) histology images are prepared from which type of tissue, spleen? Kindly indicated clearly.

4)       In section 4.7 the authors state “For platelet and TMEV antigen detection, the sections were pretreated with a 10 mM citrate..” can the authors be specific with regard to what “sections” refer to? The reader may be confused thinking that it refers to platelets instead of CNS or heart tissue sections.
